# Comparison of Korean School Students’ Safety Accident Rates before and after COVID-19

**DOI:** 10.3390/healthcare11162326

**Published:** 2023-08-17

**Authors:** Yongsuk Seo, Hyun-Su Youn

**Affiliations:** 1Sports AIX Graduate Program, Pohang University of Science and Technology, Pohang 37673, Republic of Korea; yseokss@postech.ac.kr; 2Department of Physical Education, College of Education, WonKwang University, Iksan-si 54538, Republic of Korea

**Keywords:** COVID-19 pandemic, safety accident rate, types of classes

## Abstract

The COVID-19 pandemic has significantly affected various aspects of education, including the occurrence of injuries among Korean students. This study aims to analyze and compare injury rates in elementary, middle, and high schools before and after the pandemic and identify the associated factors. A non-experimental quantitative dataset compiled from the Korea School Safety Association’s annual reports (2018–2022) was utilized. The data included information on school safety accidents among Korean children and adolescents during the COVID-19 pandemic. The dataset was analyzed based on factors such as time, location, type of accident, and injured body part. The findings revealed a decline in accidents during the early phase of the pandemic, followed by an increase after schools reopened. There were notable variations in the accidents in specific locations, types, and body parts affected during the pandemic, compared with the pre-pandemic period. This study highlights the importance of continuous monitoring, implementation of safety measures, and prioritization of physical activity programs and safety education to ensure a safe learning environment. Further research is recommended to track and address evolving school accidents in response to the pandemic and its aftermath.

## 1. Introduction

The COVID-19 pandemic has brought about unprecedented challenges and disruptions globally, significantly impacting the education sector as well [1,2]. In response to the pandemic, educational institutions have implemented a wide range of measures to ensure student safety and mitigate the spread of the virus [3]. To adapt to the new circumstances, schools and universities have adopted various strategies, such as remote learning, hybrid learning, and intermittent school closures [2,3].

With the implementation of measures such as remote learning, intermittent school closures, and changes in school environments, it is crucial to examine the effects of the pandemic on different aspects of school life, including the occurrence of injuries in Korean elementary, middle, and high schools. Several studies have investigated the impact of the pandemic on school safety and injury rates. Injuries in schools can have various causes, including accidents during physical activities, conflicts among students, and other safety-related accidents [4]. Pandemic-related measures, such as remote learning and changes in the school environment, have introduced new dynamics that may have influenced the occurrence and nature of injuries among students [5]. For instance, the shift to remote learning has altered the physical setting in which students typically engage in educational activities, potentially reducing the risk of certain types of injuries, such as those related to physical education (PE) or playground accidents. However, the increased use of digital platforms and technology for remote learning may introduce new risks or challenges that could contribute to different patterns of injury [5].

Intermittent school closures and their subsequent reopening have affected students’ daily routines, social interactions, and physical environments [6]. These changes can influence the nature of injuries as students adapt to new learning environments and face different social dynamics within school settings [7,8]. There is a close relationship between human behavior and injury patterns [9,10,11].

The implementation of lockdowns and related restrictions during epidemics, such as the COVID-19 pandemic, leads to substantial alterations in people’s daily routines and activities. These changes in behavior can have a direct impact on the occurrence and nature of injuries [12]. For instance, with reduced mobility and limited access to public spaces, certain types of injuries, such as those related to traffic accidents or recreational activities, may decline owing to decreased exposure and engagement in these activities [12]. Understanding the effects of the pandemic on injury occurrence in schools is crucial for several reasons. First, this helps to identify potential areas of concern and risks that students may face during these exceptional circumstances. Second, this provides insights into the effectiveness of the measures implemented to ensure student safety and well-being. The examination of the trends and patterns of injury before and after the pandemic permits the assessment of the effectiveness of preventive measures and the identification of areas that may require further attention or improvement. Studying the occurrence of injuries can contribute to the development of evidence-based strategies and interventions for injury prevention in schools. By identifying the factors associated with injuries during the pandemic, educators, policymakers, and stakeholders can develop targeted interventions to promote safe and secure learning environments for students.

As a part of the importance of promoting overall health, the U.S. Department of Health and Human Services released “The Physical Activity Guidelines for Americans” in 2018 [13], which provides physical activity guidelines for various age groups. For children and adolescents aged 6–17 years, it recommends at least one hour or more of moderate to vigorous physical activity each day, with the inclusion of daily or at least three days of physical activities per week. This recommendation highlights the importance of regular physical activity for children’s and adolescents’ health. Similarly, in Korea, previous studies have reported physical activity guidelines for adolescents [14,15,16]. However, it appears that the COVID-19 pandemic has severely restricted participation in physical activities by children and adolescents. Moreover, given the significant impact of the pandemic on physical activity among children and adolescents, we monitored the health status of this age group in our recent studies [17,18]. In a related study, we discovered that low participation in physical activity persisted among young individuals across six health indicators. Based on the premise that physical activity frequency, intensity, and duration remained low during the approximately three-year pandemic period, we hypothesized that insufficient physical activity among children and adolescents may negatively affect their muscles, bones, and ligaments. To test this hypothesis, we examined data from the School Health Index Conference on public data related to school safety and health. Our goal was to verify the potential impacts of the pandemic on the physical activity levels and overall health of children and adolescents in the context of their muscles, bones, and ligaments.

Examining injury rates in Korean schools is very important because several unique aspects of the Korean education system precipitate injury occurrence. For instance, the Korean education system is renowned for its high academic pressure and competitiveness, which may influence students’ behavior and vulnerability to injuries. Additionally, Korean students often spend long hours studying, which may lead to potential physical strain and exhaustion and raise a concern for their overall safety [19]. The prevalence of commuting to school over long distances further exposes students to transportation-related injuries. Moreover, the high student density and large class sizes in Korean schools may prompt accidents and injury occurrences during school hours. The COVID-19 pandemic has caused unprecedented disruptions, leading to the swift implementation of measures like remote learning and intermittent school closures in Korean schools. Understanding how these pandemic-related measures have influenced injury rates is crucial in developing targeted interventions to ensure student safety during such exceptional circumstances. Furthermore, the examination of injury patterns before and after the pandemic can provide valuable insights into the effectiveness of preventive measures and identify areas that may require further attention or improvement. This research will offer a comprehensive understanding of the interplay between the unique aspects of the Korean education system, pandemic-related measures, and injury occurrence in schools. Our findings will not only enhance school safety in Korea but also provide valuable lessons for other countries facing similar challenges in maintaining a safe learning environment for students during pandemics or other similar situations. Therefore, this study aims to analyze and compare the injury rates in Korean elementary, middle, and high schools before and after the COVID-19 pandemic. Beyond this, it seeks to identify vulnerable groups, uncover contextual factors influencing injury rates, assess the role of public health measures during the pandemic, highlight long-term implications on student safety, and provide effective injury prevention strategies for educational settings. By examining the trends and patterns, we seek to identify the broader implications of the COVID-19 pandemic on school-related injuries in Korea, offering valuable insights for improving safety measures and policies in educational institutions.

## 2. Materials and Methods

### Data Collection and Analysis

This study utilized a non-experimental quantitative dataset obtained from the annual reports of the Korea School Safety Association’s file library, covering the period from 2018 to 2022, to examine the occurrence of school safety accidents among children and adolescents in Korea. The dataset provided crucial information on the number of accidents based on various factors such as time, location, type, and body part affected.

In South Korea, an institution called the “School Safety Guarantee Association” operates the School Safety Accident Compensation Insurance program. This program aims to provide prompt and appropriate compensation to students, school staff, and education activity participants who suffer damages due to school safety accidents, based on Article 11 of the “Law on School Safety Accident Prevention and Compensation”. All participants in kindergartens, elementary, middle, and high schools, as well as educational facilities for lifelong learning, are mandatory subscribers to this insurance. Additionally, members of foreign schools under Article 60-2 of the “Elementary and Secondary Education Act” may voluntarily subscribe to this insurance. The beneficiaries of this insurance include students, school staff, and education activity participants, and the compensation covers all life and bodily injury accidents that occur during educational activities. The procedure for handling accidents after their occurrence involves the school’s prompt reporting of the accident to the local safety guarantee association through the Accident Control System (schoolsafe.or.kr) (accessed on 8 May 2023). Subsequently, the school principal or parents submit a claim form and various supporting documents (e.g., medical expense receipts and medical certificates) related to the accident. The safety guarantee association then provides compensation and notifies the claimant of the results. Through this system, information about accidents occurring in South Korean kindergartens and elementary, middle, and high schools is all recorded in a data format. The collected data, as per the Information Disclosure Act, are published on the official website of the Central School Safety Guarantee Association on an annual basis.

To analyze the data, relevant information was extracted and aligned with the objectives of the study. The collected data were then preprocessed and restructured by incorporating additional variables such as the year, pandemic phase, and form of school class. This restructuring facilitated a comprehensive understanding of the relationships between these factors and the occurrence of accidents. The resulting dataset was compiled in a table format, with the frequency and percentage of accidents in each category.

This dataset serves as a valuable resource for analyzing and comprehending the patterns and characteristics of school safety accidents in educational settings. The incidence of injuries was categorized based on factors such as time, location, cause, and specific body parts affected. By examining these variables, a comprehensive understanding of the temporal distribution of injuries, the settings in which they occurred, the contributing causes, and the commonly affected body parts was obtained.

The analysis of this dataset provides valuable insights into the occurrence of school safety accidents during the pandemic. It offers a deeper understanding of how accidents are distributed over time, the specific settings where they are more likely to occur, the factors that contribute to their incidence, and the body parts that are frequently affected. These insights can aid policymakers, educators, and parents in developing and implementing targeted safety measures to reduce the occurrence of accidents and promote a safer learning environment for children and adolescents during challenging times.

## 3. Results

From 2018 to 2022, a range of accidents involving students from kindergarten to high school level occurred in Korea. These accidents encompassed various types, including transportation accidents, falls, sports-related injuries, and other incidents occurring within or outside the school premises.

An analysis of the accident data, as shown in Table 1, reveals some notable trends before and during the COVID-19 pandemic. Prior to the pandemic, during the years 2018 and 2019, the number of accidents remained relatively consistent. This suggests a stable pattern in the occurrence of accidents among students during that period. However, with the onset of the COVID-19 pandemic in 2020, there was a sharp decline in the number of reported accidents. This decline can be attributed to the significant changes in school environments and activities resulting from the implementation of preventive measures, such as school closures, remote learning, and reduced physical interactions.

In 2021, there was a noticeable increase in the number of accidents compared to the previous year. This rise can be attributed to several factors. As schools gradually resumed in-person classes or adopted hybrid learning models, students re-entered the physical school environment after an extended period of remote learning. Adjusting to the changes and re-establishing routines may have contributed to an increased number of accidents as students readapted to their school surroundings. Furthermore, the increased number of accidents in 2022 compared to the pre-pandemic period suggests a continuing shift in the accident patterns. It is important to note that the reasons behind this increase could be multifaceted. Factors such as the relaxation of certain preventive measures, changes in student behavior or supervision, or other external factors may have contributed to this trend.

Table 2 and Figure 1 categorize yearly trends in school accident occurrence by the time of day. Before the COVID-19 pandemic in 2018 and 2019, the highest number of accidents occurred during PE classes, followed by lunchtime, regular class hours, and break/cleaning time. This could be attributed to the nature of PE activities, which often involve physical exertion and a higher risk of injury.

During the pandemic, in 2020, there was a decrease in the overall number of accidents across all periods compared with the pre-pandemic period. This decline can be attributed to various factors, including reduced student presence on school premises due to remote learning, limitations on extracurricular activities, and stricter adherence to safety protocols. However, in 2021, there was an increasing trend in the number of accidents. In 2022, there was an increase in accidents during PE classes compared with the pre-pandemic period, while other periods showed a similar pattern.

Yearly trends in school accident occurrences were categorized by location, as shown in Table 3 and Figure 2. Before the COVID-19 pandemic, in 2018 and 2019, the highest number of accidents occurred on playgrounds, followed by auxiliary facilities, classrooms, hallways, and off-campus activity areas. In the first year of the pandemic, there was a decrease in the occurrence of accidents in all locations compared with the pre-pandemic period.

However, in 2021, there was an increasing trend in the number of accidents compared with the previous year. In 2022, the data show a notable increase in accidents on the playground and in auxiliary facilities compared with the pre-pandemic period. In 2022, there was an increase in accidents on the playground and in auxiliary facilities, whereas accidents in other locations remained at a constant level, compared with the pre-pandemic period. This indicates that as schools returned to more regular operations, the risks associated with outdoor play and the use of auxiliary facilities resurfaced. However, the accidents reported in other locations, such as classrooms and hallways, remained relatively constant compared with the pre-pandemic period.

Table 4 and Figure 3 present the annual trends in school accidents categorized by type. Before the onset of the COVID-19 pandemic, in 2018 and 2019, the most common type of accident was “exposure to physical force”. This was followed by “falls (tripping)”, “collisions with people”, “falls (slipping)”, “other accidents”, and “falls (falling)”. During the COVID-19 pandemic, in 2020, all types of accidents decreased compared with the pre-pandemic period. However, in 2021, the number of accidents of all types increased. In 2022, the types of accidents that saw an increase compared with the pre-pandemic period were “exposure to physical force”, “falls (tripping)”, and “falls (slipping)”. Other accident types exhibited similar trends. Notably, collisions with people exhibited a significant decrease compared with the pre-pandemic period.

Table 5 and Figure 4 present the annual trends in school accidents categorized by body part. Before the COVID-19 pandemic, in 2018 and 2019, the body parts with the highest number of accidents were the hands and feet, followed by the head. This was followed by the legs, arms, oral cavity, chest, abdomen, and other body parts.

During the COVID-19 pandemic, in 2020, all body parts, except for “other”, experienced a decrease in the number of accidents compared with the pre-pandemic period. However, in 2021, the number of accidents involving all body parts increased. This upward trend indicates that as schools began to resume in-person classes or adopt hybrid learning models, the number of accidents involving different body parts started to rise again. The reasons behind this increase may vary and could be influenced by factors such as students’ readjustment to physical activities, changes in supervision, or other external factors. In 2022, the number of accidents involving the hands and feet increased compared with the pre-pandemic period, whereas other body parts exhibited a decrease in the number of accidents.

## 4. Discussion

During the COVID-19 pandemic, elementary, middle, and high schools in Korea implemented various measures to minimize the risk of COVID-19 transmission among students [13]. Various forms of classroom instruction during the COVID-19 pandemic resulted in diverse patterns of school safety accidents. In 2020, schools implemented full-scale remote learning in which all classes were conducted online. This was in response to the COVID-19 pandemic in order to ensure the safety of students and prevent the spread of the virus [20]. In 2021, a hybrid model for remote and in-person learning was adopted. This involved a combination of online classes and limited in-person instruction [21]. Schools implemented measures such as alternating schedules or dividing students into smaller groups to maintain social distancing and reduce the risk of COVID-19 transmission [20]. The different instructional formats adopted by schools affected the occurrence and nature of school accidents.

The main findings of this study are as follows: (1) The number of safety accidents in schools regressed during the pre-pandemic period. (2) Accidents increased during PE classes. (3) The number of accidents in sports fields and related facilities increased compared with the pre-pandemic period [5]. Specific types of accidents, such as exposure to physical force, falls, and slips showed an increase compared with the pre-pandemic period.

Initially, in 2022, when educational institutions transitioned back to in-person classes and resumed most educational activities in keeping with pre-pandemic norms, there was a notable regression in the number of school safety accidents. This regression can be attributed to several factors associated with the return to a more familiar learning environment and the resumption of face-to-face instruction. The change in instructional formats played a significant role in the observed increase in safety accidents. During the pandemic, schools relied heavily on remote learning and hybrid models, where students had limited in-person interaction and were more likely to be in controlled environments at home. However, with the return to traditional in-person classes, students were exposed to a different set of risks and challenges associated with physical proximity to others, such as accidental falls, collisions, or other incidents that can occur within school premises.

Moreover, the restoration of regular school routines after a prolonged period of disruption could have also contributed to the increase in accidents. The transition back to pre-pandemic norms may have caused some students to feel a sense of familiarity and relaxation, which could lead to a temporary decrease in vigilance regarding safety protocols. This could include behaviors like running in hallways, not paying attention while walking, or engaging in other activities that could increase the likelihood of accidents. It is important to note that the observed increase in school safety accidents after the return to normality does not suggest that in-person instruction is inherently unsafe. Rather, it reflects the need for continued vigilance, the reinforcement of safety protocols, and ongoing efforts to create a safe and secure learning environment for students. To address this issue, educational institutions should prioritize safety education and awareness programs to remind students of the importance of following safety guidelines and being cautious in their actions. Implementing preventive measures such as maintaining clear pathways, providing safety equipment, and promoting supervision and monitoring in high-risk areas can also help mitigate the occurrence of accidents.

The number of safety accidents during PE classes increased compared with those in the pre-COVID-19 period. This can be attributed to the lack of physical activity during the pandemic, as remote learning and sedentary habits became more prevalent [22]. Limited experience with physical activity may have impaired students’ physical coordination abilities, leading to a higher occurrence of safety accidents during resumed PE classes [23]. For example, Peterson and Renström [24] reported that children are especially susceptible to injuries due to various factors, including their immature reflexes, underdeveloped coordination, and limited capacity to recognize and evaluate risks [24]. While most PE injuries among children are minor, some may necessitate medical attention and result in school absences [25].

Regular physical activity plays a vital role in improving various aspects of health, including cardiovascular and muscular fitness, bone health, psychological well-being, cognitive function, and brain health [2]. By incorporating regular physical activity into one’s lifestyle, individuals can significantly reduce the risk of developing chronic diseases, such as cardiovascular diseases, type 2 diabetes, certain types of cancer, and obesity [26]. Moreover, regular physical activity has been shown to increase longevity and lower the risk of premature mortality [26]. More importantly, previous research has indicated that developing a habit of physical activity during childhood is closely associated with the continuation of physical activity habits in later life stages [27]. Furthermore, regular physical activity plays a crucial role in maintaining physical fitness, coordination, and overall well-being [28]. Research has shown that physical activity during childhood is correlated with adopting healthier lifestyles in adulthood. The decrease in physical activity during the pandemic may have led to a decline in students’ physical abilities and preparedness for PE classes, potentially contributing to an increased frequency of safety accidents [29].

The number of accidents in playgrounds and auxiliary facilities (gymnasiums) increased compared with the pre-COVID-19 period. Even before the pandemic, playgrounds and auxiliary facilities had the highest number of accidents [30,31]. This trend continued in 2022, with a higher incidence of accidents in these areas, as previously mentioned. It can be inferred that the reported increase in safety accidents reflects the increase in accidents during PE classes conducted in playgrounds and auxiliary facilities (gymnasiums) [32]. It is important to note that homes and schools are the most common locations where children can sustain injuries. According to statistics, approximately 26% of all reported accidents during compulsory school attendance occur in the school environment [32]. Playgrounds and auxiliary facilities are often associated with various physical activities including sports, games, and recreational exercise. These activities involve high levels of movement and physical exertion, which may increase the risk of accidents and injuries [30,31]. Schools should pay attention to safety measures and supervision in these areas to prevent accidents and ensure students’ well-being.

The types of accidents that increased during the COVID-19 pandemic included physical force exposure, falls, and slipping. These types of accidents showed an upward trend compared with the pre-COVID-19 period. The increase in accidents related to exposure to physical force, falls, and slipping can be attributed to several factors. During the pandemic, changes in the learning environment and restrictions on physical activity may have led to reduced physical fitness and coordination among students [29].

The lack of regular physical exercise and decreased opportunities for movement and outdoor activities may have contributed to the higher incidence of accidents related to physical force and falls. Furthermore, the number of safety accidents involving the hands and feet increased during the COVID-19 pandemic compared with the pre-pandemic period. These accidents often manifest as minor injuries to the wrists or ankles, which are considered to be smaller areas of the body. This increase in the number of accidents can be attributed to several factors. During the pandemic, physical inactivity and sedentary lifestyles may have contributed to weakened muscles, bones, and ligaments, making people more susceptible to injuries in these smaller areas [33]. Additionally, factors such as being overweight or obese can further increase the risk of accidents and injuries to the hands and feet [33]. In contrast, accidents involving other body parts, such as the head, oral cavity, chest, legs, and arms, decreased during the pandemic. These body parts are likely to be involved in accidents or injuries resulting from interpersonal collisions or contact. Overall, understanding the specific patterns and causes of accidents related to different body parts can guide schools in implementing targeted preventive measures and safety education programs. By addressing the underlying factors contributing to these accidents, schools can reduce accidents and promote the overall well-being of their students.

During the COVID-19 pandemic, factors such as changes in school policies, parental supervision, and environmental conditions played significant roles in shaping the safety of children and adolescents, which is in agreement with previous studies, indicating that altering school policies and recreational activities to adhere to social distancing guidelines may have influenced accident rates among students [34]. Variations in parental supervision during lockdowns and remote learning could impact accidents outside of school. The availability of safe outdoor spaces, healthcare facilities, and changes in traffic patterns might have also influenced injury rates [35]. Considering these factors allows for a more comprehensive analysis, thus aiding in identifying strategies to address safety concerns during exceptional circumstances, like the COVID-19 pandemic. Furthermore, previous studies reported reduced physical activity during the pandemic, attributing this to restrictive measures such as school closures and limited outdoor activities [1]. Additionally, the potential impact of the pandemic on students’ mental health and well-being is consistent [36,37].

This study has limitations that should be considered for generalization and interpretation. The current study’s primary focus on descriptive analysis poses a limitation as it may hinder the exploration of underlying causes and mechanisms behind the observed changes in injury rates. A more comprehensive approach involving deeper investigation would help in gaining a more profound understanding. Moreover, the study heavily relies on retrospective data from annual reports, which, while providing a substantial amount of information, could be susceptible to reporting biases or incompleteness. Incorporating additional data sources or employing a prospective study design could help mitigate these potential limitations and strengthen the study’s overall credibility. By doing so, the research outcomes would be more robust and reliable, allowing for a more comprehensive analysis of the factors influencing injury rates. Limited existing literature on the long-term implications of the pandemic for school safety accidents and physical activity underscores the importance of the study’s contribution in examining the effects over a significant time frame and providing valuable insights into the potential enduring impacts on students’ well-being. To gain a comprehensive understanding of school injuries and their evolution over time, future research should adopt a multifaceted approach, such as longitudinal data, cross-cultural comparisons, in-depth case studies, and comprehensive data collection. Additionally, exploring the influence of student demographics, behavior, technology use, school policies, and community involvement would inform targeted and adaptable safety interventions for creating safer learning environments.

The present study can provide practical implications for informing school policies and safety measures during pandemics and beyond. First, adapting the PE curriculum to prioritize regular physical activity, even during remote learning or intermittent school closures, could help counteract the potential decline in activity levels observed during the pandemic. Second, enhancing online safety education would equip students with the knowledge to navigate digital platforms responsibly, considering the increased reliance on technology for remote learning. Prioritizing mental health support and counseling services can address the potential impact of the pandemic on students’ mental well-being, fostering emotional resilience and reducing the likelihood of risky behaviors that lead to accidents. Strengthening communication between schools and parents would facilitate collaborative efforts to ensure student safety during remote learning periods, given the significant role parental supervision may play. Integrating targeted injury prevention programs based on the study’s insights can contribute to reducing accident rates among students. Comprehensive emergency preparedness planning and addressing safety concerns during pandemics and other crises will be essential for the safety and well-being of students. By incorporating these implications, educational institutions can create a holistic approach to prioritizing student well-being and foster positive learning experiences under various circumstances. Further research is needed to explore the specific factors that contribute to the observed changes in injury rates. Examining the types and causes of accidents as well as the specific body parts affected can provide valuable insights for targeted interventions and preventive measures. Additionally, longitudinal studies that track injury rates over extended periods and compare them across different regions could enhance our understanding of the long-term effects of pandemics on school safety.

## 5. Conclusions

By examining the overall accident status of elementary, middle, and high school students, policymakers, educators, and stakeholders can identify areas of concern and develop effective strategies and preventive measures to enhance student safety and well-being in Korean schools. It is crucial to monitor and address accident trends and patterns continuously to ensure a safe and secure learning environment for all students. The initial decline in accidents during the early phase of the pandemic was followed by an increase after the reopening of schools. These findings emphasize the need for continuous monitoring and implementation of appropriate safety measures to develop and support programs aimed at restoring and enhancing physical activity in children and adolescents. Additionally, the systematic monitoring of students by PE teachers, along with the thorough implementation of accident prevention education for all school members, is necessary. Future research should track and observe the evolving nature of accidents in schools.

## Figures and Tables

**Figure 1 healthcare-11-02326-f001:**
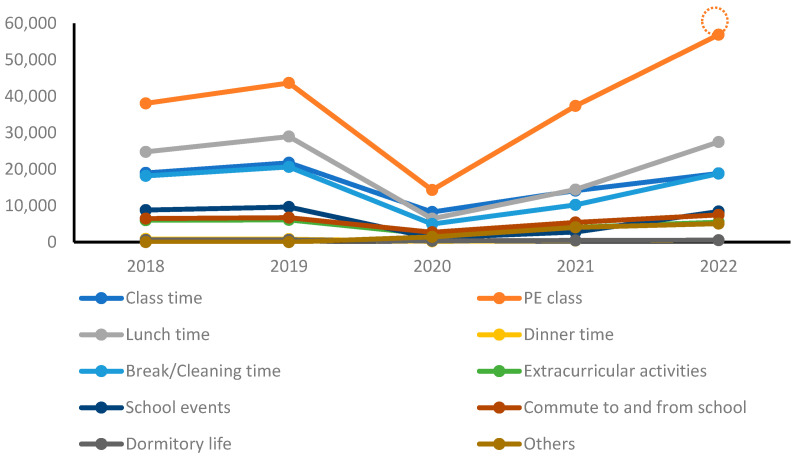
Yearly trends of school accidents by time of day.

**Figure 2 healthcare-11-02326-f002:**
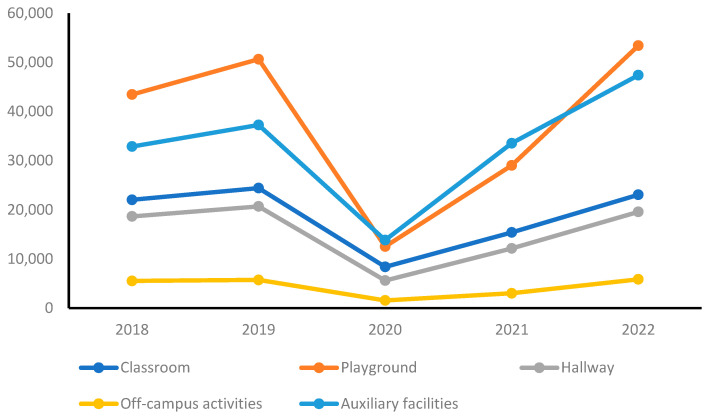
Yearly trends of school accidents by location.

**Figure 3 healthcare-11-02326-f003:**
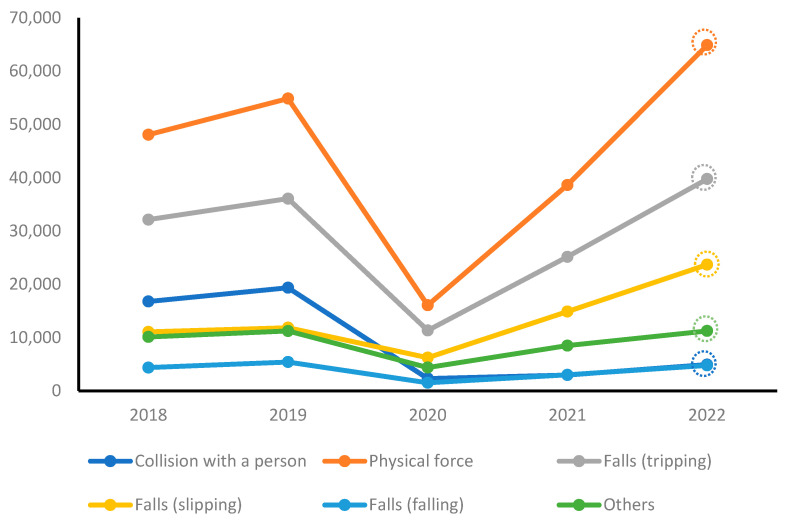
Yearly trends of school accidents by type.

**Figure 4 healthcare-11-02326-f004:**
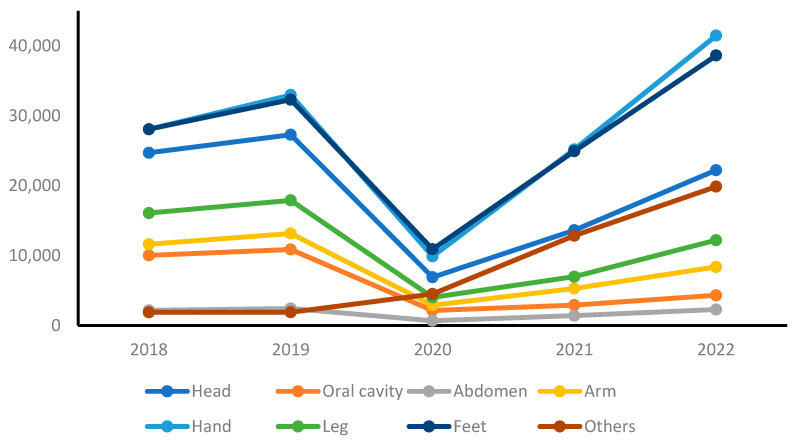
Yearly trends of school accidents by injured body part.

**Table 1 healthcare-11-02326-t001:** Yearly trends of overall school accidents.

Year		Prior to COVID-19 Pandemic		During COVID-19 Pandemic		
		2018	2019	2020	2021	2022
Types of Classes		In Person		Blended Learning		In Person
Total	Cases	122,570	138,784	41,940	93,147	149,339

**Table 2 healthcare-11-02326-t002:** Yearly trends of school accidents categorized by time of day.

Year		Prior to COVID-19 Pandemic		During COVID-19 Pandemic		
		2018	2019	2020	2021	2022
Types of Classes		In Person	Remote	Remote	Blended	In Person
Class time	Cases	18,963	21,746	8246	14,109	18,828
	%	15.47%	15.67%	19.66%	15.15%	12.6%
PE class	Cases	38,031	43,619	14,274	37,318	56,841
	%	31.03%	31.43%	34.03%	40.06%	38.1%
Lunch time	Cases	24,741	28,936	6440	14,390	27,441
	%	20.19%	20.85%	15.36%	15.45%	18.4%
Dinner time	Cases	875	823	223	417	528
	%	0.71%	0.59%	0.53%	0.45%	0.4%
Break/Cleaning time	Cases	18,151	20,620	4969	10,201	18,775
	%	14.81%	14.86%	11.85%	10.95%	12.6%
Extracurricular activities	Cases	5984	6111	2125	4057	5470
	%	4.88%	4.40%	5.07%	4.36%	3.7%
School events	Cases	8776	9615	1169	2766	8395
	%	7.16%	6.93%	2.79%	2.97%	5.6%
Commute to and from school	Cases	6445	6706	2720	5385	7466
	%	5.26%	4.83%	6.49%	5.78%	5.0%
Dormitory life	Cases	604	608	347	464	522
	%	0.49%	0.44%	0.83%	0.50%	0.3%
Others	Cases	0	0	1427	4040	5073
	%	0.00%	0.00%	3.40%	4.34%	3.4%
Total	Cases	122,570	138,784	41,940	93,147	149,339
	%	100%	100%	100%	100%	100%

**Table 3 healthcare-11-02326-t003:** Yearly trends in school accidents by location.

Year		Prior to COVID-19 Pandemic		During COVID-19 Pandemic		
		2018	2019	2020	2021	2022
Types of Classes		In Person	Remote	Remote	Blended	In Person
Classroom	Cases	22,040	24,434	8401	15,401	23,072
	%	18.0%	17.6%	20.0%	16.5%	15.4%
Playground	Cases	43,466	50,653	12,541	29,043	53,419
	%	35.5%	36.5%	29.9%	31.2%	35.8%
Hallway	Cases	18,660	20,692	5607	12,144	19,585
	%	15.2%	14.9%	13.4%	13.0%	13.1%
Off-campus activities	Cases	5525	5734	1558	3011	5854
	%	4.5%	4.1%	3.7%	3.2%	3.9%
Auxiliary facilities	Cases	32,879	37,271	13,833	33,548	47,409
	%	26.8%	26.9%	33.0%	36.0%	31.7%
Total	Cases	122,570	138,784	41,940	93,147	149,339
	%	100%	100%	100%	100%	100%

**Table 4 healthcare-11-02326-t004:** Yearly trends of school accidents by type.

Year		Prior to COVID-19 Pandemic		During COVID-19 Pandemic		
		2018	2019	2020	2021	2022
Types of Classes		In Person	Remote	Remote	Blended	In Person
Collision with a person	Cases	16,777	19,355	2319	2977	4933
	%	13.7%	13.9%	5.5%	3.2%	3.3%
Physical force	Cases	48,073	54,844	16,089	38,622	64,884
	%	39.2%	39.5%	38.4%	41.5%	43.4%
Falls (tripping)	Cases	32,140	36,075	11,360	25,149	39,778
	%	26.2%	26.0%	27.1%	27.0%	26.6%
Falls (slipping)	Cases	11,080	11,865	6248	14,883	23,701
	%	9.0%	8.5%	14.9%	16.0%	15.9%
Falls (falling)	Cases	4380	5412	1529	3021	4783
	%	3.6%	3.9%	3.6%	3.2%	3.2%
Others	Cases	10,120	11,233	4395	8495	11,260
	%	8.3%	8.1%	10.5%	9.1%	7.5%
Total	Cases	122,570	138,784	41,940	93,147	149,339
	%	100%	100%	100%	100%	100%

**Table 5 healthcare-11-02326-t005:** Yearly trends of school accidents by body part.

Year		Prior to COVID-19 Pandemic		During COVID-19 Pandemic		
		2018	2019	2020	2021	2022
Types of Classes		In Person	Remote	Blended Learning		In Person
Head	Cases	24,694	27,278	6893	13,619	22,215
	%	20.1%	19.7%	16.4%	14.6%	14.9%
Oral cavity	Cases	10,028	10,868	2146	2911	4312
	%	8.2%	7.8%	5.1%	3.1%	2.9%
Abdomen	Cases	2159	2426	680	1398	2270
	%	1.8%	1.7%	1.6%	1.5%	1.5%
Arm	Cases	11,604	13,155	2897	5283	8370
	%	9.5%	9.5%	6.9%	5.7%	5.6%
Hand	Cases	28,048	32,969	9894	25,183	41,472
	%	22.9%	23.8%	23.6%	27.0%	27.8%
Leg	Cases	16,080	17,900	4020	6981	12,202
	%	13.1%	12.9%	9.6%	7.5%	8.2%
Feet	Cases	28,072	32,299	10,913	24,927	38,629
	%	22.9%	23.3%	26.0%	26.8%	25.9%
Others	Cases	1885	1889	4497	12,845	19,869
	%	1.5%	1.4%	10.7%	13.8%	13.30%
Total	Cases	122,570	138,784	41,940	93,147	149,339
	%	100%	100%	100.0%	100%	100.0%

## Data Availability

Publicly available datasets were analyzed in this study. These data can be found here: https://www.schoolsafe.or.kr/school/login.do (accessed on 8 May 2023).

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
