# Peer review of "Comparison of Korean School Students’ Safety Accident Rates before and after COVID-19"

_healthcare, 2023, doi:10.3390/healthcare11162326_

Round 1
Reviewer 1 Report
The study used a quantitative dataset compiled from the annual reports of the Korea School Safety Association, covering a five-year period from 2018 to 2022. This allowed for a comprehensive and comparative analysis of injury rates in different levels of schooling before and after the pandemic. The use of comprehensive data enhances the reliability and validity of the obtained results. By considering multiple factors, the study provides more comprehensive insights into risks and priority areas for implementing safety measures and educational programs.
One limitation of the study being predominantly descriptive is that it may lack a deeper understanding of the underlying causes and mechanisms driving the observed changes in injury rates. Another limitation is the reliance on retrospective data from annual reports. While these reports provide a wealth of information, they may be subject to limitations such as potential reporting biases or incomplete data.
Despite its descriptive nature, the study still offers valuable information that contributes to our knowledge of the phenomenon by identifying trends, raising awareness of potential risks, and guiding the development of future research and safety measures.
Reviewer 2 Report
Introduction
The introduction lacks a clear statement of the specific objectives of the study. It would be helpful to explicitly state what the study aims to achieve beyond analyzing and comparing injury rates.
Literature Review: While the introduction mentions that several studies have investigated the impact of the pandemic on school safety and injury rates, it lacks a more comprehensive literature review that cites and discusses these previous studies.
Rationale for Study: The introduction should provide a stronger rationale for conducting this research. Why is it important to examine injury rates in Korean schools specifically? Are there any unique aspects of the Korean education system that make this study relevant and significant?
It would be beneficial to establish a conceptual framework or theoretical basis for the study. Are there any existing theoretical models or frameworks that can guide the exploration of factors influencing injury rates during the pandemic?
Methods
The section mentions that the data were obtained from the annual reports of the Korea School Safety Association's file library. How was the data collected, and what quality control measures were taken to ensure accuracy and consistency?
Also, data collection procedures should be disclosed.
The section mentions that the collected data were preprocessed and restructured to incorporate additional variables. Provide more details on the specific preprocessing steps and how missing data or outliers were handled. Transparently reporting data preprocessing steps helps ensure the study's reproducibility.
Describe the statistical methods used to analyze the dataset.
If the data involves sensitive information or involves human participants, discuss the ethical considerations taken to protect their privacy and adhere to ethical guidelines.
Results
It is essential to include statistical analyses to support the reported trends and differences in accident occurrences. Utilize appropriate statistical tests to determine the significance of observed changes. Provide confidence intervals to quantify the uncertainty in the estimates.
Clarify whether the dataset used is a representative sample of all schools in Korea. Address any potential biases or limitations in the dataset's representativeness, which could affect the generalizability of the findings.
Discussion
While the study identifies associations between the pandemic and changes in school safety accidents, it is crucial to emphasize the limitations of establishing causation from observational data.
The discussion highlights factors like reduced physical activity during the pandemic that might have contributed to the increase in certain accidents. However, the discussion would benefit from considering other potential confounding variables, such as changes in school policies, parental supervision, or environmental factors.
To contribute to practical implications, discuss how the study's findings can inform school policies and safety measures during pandemics and beyond.
Compare the study's findings with existing literature on school safety, accidents, and physical activity. Identify areas of agreement or discrepancy and discuss possible explanations for any inconsistencies.
The discussion should include a clear section on study limitations, acknowledging potential biases and uncertainties.
The overall quality of English in the paper is good, and the text is generally understandable. However, there are some areas where minor improvements are needed to enhance clarity and fluency. Some sentences could be rephrased to improve readability, and a few grammar and punctuation errors should be corrected.
Reviewer 3 Report
First, I would like to congratulate the authors for this interesting work. They present an article with interesting and detailed information. I would like to comment on some considerations, which I believe will contribute to improve the final presentation.
The introduction is too short. They should expand on the information about injuries and how they have evolved worldwide over the years, so that the reader can understand the phenomenon. It is only clear why it is important to analyse injuries.
In the section Material and Methods much information is missing. Only "annual reports from the Korean School Safety Association's archive library" are mentioned. Don't these reports have a bibliographic reference?
They do not mention the participants in these reports, how the data collection is carried out, their procedure, what criteria are used to classify the types of injuries,... These are indispensable elements in an study.
Legends of figures and tables should be written appropriately.
The results are presented in detail, but are discussed. Each piece of information should go in its corresponding section. In the results only the results should be described, and in the discussion the results are discussed, relating them to previous findings in the field.
The limitations of the study are not provided and practical applications need to be further explored.
Round 2
Reviewer 3 Report
Good morning.
The text is much improved compared to the previous version.
I think it can be published.
Best regards